# A Low-Cost Contactless Overhead Micrometer Surface Scanner

Xenophon Zabulis *⬤, Panagiotis Koutlemanis ⬤, Nikolaos Stivaktakis ⬤ and Nikolaos Partarakis ⬤

Institute of Computer Science, Foundation for Research and Technology (ICS-FORTH), N. Plastira 100, Vassilika Vouton, GR-700 13 Heraklion, Greece; koutle@ics.forth.gr (P.K.); nstivaktak@ics.forth.gr (N.S.); partarak@ics.forth.gr (N.P.)
* Correspondence: zabulis@ics.forth.gr

**Featured Application: Contactless and high-resolution scanning of planar surfaces find application in several disciplines that involve sensitive or deformable samples, such as the study and conservation of heritage objects, textiles, documents, and paintings. Off-the-shelf solutions come at a high cost. In this work, an approach to contactless, high-resolution scanning of planar surfaces is proposed that is relevant to several applications. The implementation of this approach exhibits reduced hardware cost, is based on open designs, and can be replicated with ease.**

**Abstract:** The design and implementation of a contactless scanner and its software are proposed. The scanner regards the photographic digitization of planar and approximately planar surfaces and is proposed as a cost-efficient alternative to off-the-shelf solutions. The result is 19.8 Kppi micrometer scans, in the service of several applications. Accurate surface mosaics are obtained based on a novel image acquisition and image registration approach that actively seeks registration cues by acquiring auxiliary images and fusing proprioceptive data in correspondence and registration tasks. The device and operating software are explained, provided as an open prototype, and evaluated qualitatively and quantitatively.

**Keywords:** scanner; imaging, image registration; image mosaic; image stitching; 3D printer

## 1. Introduction

The generation of image mosaics out of partial images of a surface is a useful task in many applications. Mosaics are useful because they image a larger amount of surface than a single image does. If a mapping between pixel and metric coordinates is achieved, then world measurements can be performed using the mosaic, much like in cartographic maps. Image registration upon general surfaces enables photorealistic maps for Geographical Information Systems; photopanoramas [1]; and also specialized mosaics from usually unseen surfaces such as the inner of pipes [2], the gastrointestinal tract [3], and the human retina [4]. At the core of all methods for image mosaic generation is the problem of image registration.

The simplest case of mosaic generation is met when imaging a planar surface, by multiple and conveniently tessellated overlapping frontal views. Just this case is useful in several domains, such as remote sensing [5], document scanning [6], bioinformatics [7,8], art [9], and others. Approaches to this problem that are based purely on visual cues are continuously making progress but, given pixel quantization, they exhibit error. For a large number of images, this error accumulates and gives rise to distortions. For this reason, applications that require large mosaics make use of independent information about the location of the camera. For example, photorealistic stitching of remote sensing and aerial images is supported by GPS measurements. We use this principle in the context of overhead scanning, where approximate location measurements are available from the motion mechanism of the scanner.

The optical resolution of scanners is measured by the number of pixels, or points, per unit area. We use the equivalent of dots per inch—that is, points per inch (ppi). The ppi

is a 1D metric that denotes the resolution of points across a line and, thus, a resolution of 100 ppi means that in the scanned image, $100 \times 100 = 10$ Kp would be devoted for a surface area of 1 in $\times$ 1 in = 1 in$^2$. It is equivalent to say that the scanned image has a resolution of 10 Kp/in$^2$ (number of points/per square inch).

In this work, a contactless, flatbed scanner, which costs less than 1000 USD, offers 19754 ppi resolution, and has a scanning area of 33 $\times$ 37 cm$^2$, is proposed. Accurate surface mosaics are obtained based on a novel image acquisition and image registration approach that actively seeks registration cues by acquiring auxiliary images and utilizing proprioceptive data in correspondence and registration tasks. The implementation in a device with accompanying software is presented, provided as an open prototype, and evaluated qualitatively and quantitatively.

## 1.1. Visual Registration of Images

The problem of computational image registration dates back at least four decades of study. Methods in the literature are usually called "local" if they use point features correspondence or "global" if they use overall image similarity [10,11]. When combining images in a mosaic, these images are required to have some lateral "overlap" and the registration task is called stitching. Due to restricted overlap, stitching is more accurate when local methods are used.

Given point correspondences across two images of a planar surface, robust registration solutions have been found and, by now, are textbook material [12,13]. When many images are registered sequentially in a mosaic, the error is accumulated, distorting the result. A solution is to employ a "global alignment" or "bundle adjustment" step, which either obtains a more accurate solution or, at least, distributes error so that the shape of the scanned area is retained. Although this improves the result, the error from the registration of many images manifests as local, noticeable distortions, often called "seams". For a few images, these distortions are small and well-treated by methods that reduce their visual prominence. However, when the number of images is increased by two orders of magnitude, we observed these distortions to become significant and noticeable at close and macroscopic inspection.

## 1.2. Proprioceptive Image Registration

Another way to register images in mosaics is employed by scanners, which use mechanisms to drive the sensor at locations where the acquired images would precisely match. The utilized sensors are most often line cameras with intense illumination and are less-often photographic cameras. A market survey of pertinent solutions can be found in Appendix A.

Contact-based, flatbed scanners provide up to 1000 ppi at a significantly high cost. Large-format scanners provide resolutions of up to 1200 ppi and are almost contactless. However, the scanned material should be less than a thickness threshold, i.e., 3 cm, to pass through the scanning slit and also exhibit high cost. Film scanners exhibit higher resolution, but require contact, material transparency, and are limited to the frame size of photographic film.

Large-format scanners are contactless and designed for sensitive documents but are also used for scanning textiles and other similar materials. They reach up to *A0* scanning size. The precision required for this mechanical task elevates the cost of the scanning hardware.

Book scanners are contactless and exhibit resolution in the range of 600–1200 ppi. Their cost ranges from low to very high, though in many cases, the elevated cost is due to the mechanics for the automation of page-turning. It ought to be noted though that V-shaped—as opposed to flatbed—book scanners are unsuitable in the case of deformable materials such as fabrics or sand.

Recently, the need for realistic textures gave rise to flatbed, contactless surface scanners, called "material scanners". They use camera photography and exhibit resolutions of up

to 1000 ppi. Their effective scanning surface is in the order of $30 \times 40$ cm$^2$. Lower-cost material scanners utilize the sensor of the mobile phone [14], the result however exhibits lower resolution and definition compared to the aforementioned solutions.

### 1.3. The Proposed Approach

To the best of our knowledge, the proposed work exhibits the following novel characteristics.

The cost-efficient motion mechanism of a Cartesian 3D printer (C3Dp) is used to (a) systematically place the camera at prescribed locations and (b) to collect proprioceptive data that improve registration accuracy. The motion model of the C3Dp provides 3 Degrees of Freedom (see Figure 1, left). These are sufficient to place the camera at any frontal posture of the volume above the scanned surface. The scanner motors may exhibit backlash, which points to localization uncertainty, but not exhibit error accumulation. Using this motion mechanism, images are systematically acquired. We compensate for the potential lack of accuracy due to low-end hardware or mechanical jitter by strategically acquiring auxiliary images that help the generation of spatially accurate scans.

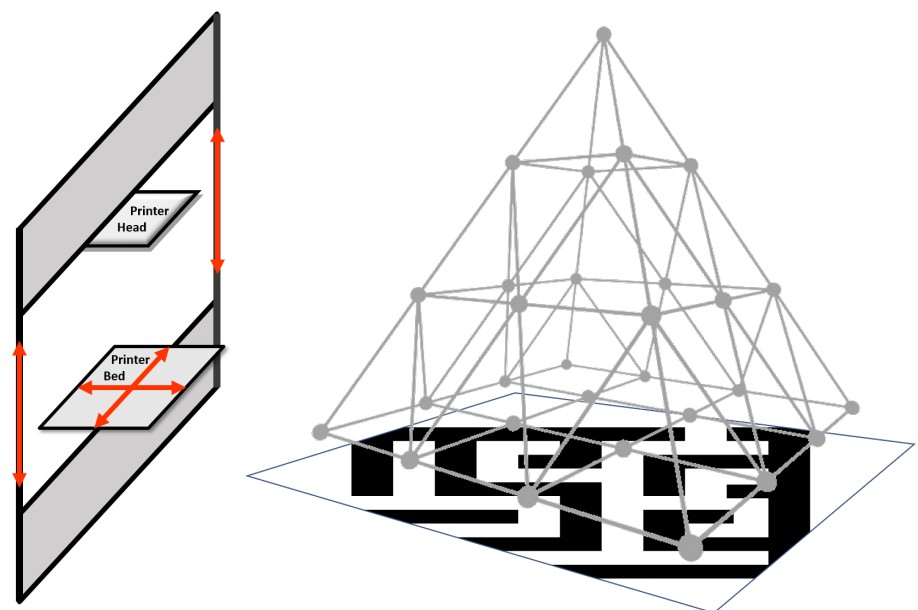

**Figure 1.** Abstraction of C3Dp motion (**left**, recreated from [15]) and proposed scanning locations (**right**).

Image registration is based on the fusion of visual and proprioceptive data. The stepper motor is used to anchor error accumulation per motor step, due to the (approximately) repeatable nature of stepper motors. To cope with the uncertainty of motor error, we strive to acquire as many registration cues as our memory can store. Images that capture neighboring surface regions overlap laterally. Please confirm for all Additionally, auxiliary images are acquired. These images overlap medially, due to elevations of the camera relative to the scanned surface, and are used to provide additional registration cues. However, the registration of images at different scales is not trivial for large-scale differences [16,17]. Therefore, we elevate the camera in controlled steps, safeguarding the preservation of registration cues across the scale.

To maximize mosaic resolution, we obtain the largest number of pixels per unit area that we can. Thus, we select an affordable lens that can provide focused images at the closest possible distance or, otherwise, a "macro lens". Telecentric lenses would be extremely more useful. However, not only they are more expensive, but they are also bigger and heavier. The latter two properties would escalate the cost of a motion mechanism with the same motion precision much more than the lens would.

As a trade-off between the available memory and the requirement for as many registration cues as possible, we acquire images in a pyramidic scheme, as shown in Figure 1 (right). The base of the pyramid (called layer 0) represents the set of images acquired as close as possible to the surface. Higher layers represent imaging at larger distances. In the figure, each node represents a point in 3D space where the sensor will acquire a frontal image of the scanned surface.

## 2. Materials

The proposed approach is implemented using the following materials.

### 2.1. Off-the-Net and Off-the-Shelf Components

The proposed Cartesian 2D scanner (C2Ds) is a device that is attached next to the printing head of a C3Dp. The C3Dp is not otherwise modified; thus, the attachment can be removed without affecting its operation. The motion mechanism belongs to the C3Dp. This mechanism moves the printing plate laterally, in two dimensions, and the printing head only vertically. The C3Dp is commanded to reach the imaging locations by a microcontroller.

The C2Ds was built on top of an adaptation of the Prusa i3 series C3Dp, which were chosen due to their wide adoption, low cost, and ease of construction. The operating volume is $24.89 \times 21.08 \times 6.86$ cm$^3$. The selected parts for the C3Dp are cited in Appendix A.

The visual sensor was an Olympus Tough TG-5, which has a minimum focus distance of 1 cm, $4000 \times 3000$ p resolution, and a FoV of $16° \times 12°$.

The motor is controlled by the Marlin open-source firmware. Marlin is widely used and runs on the cost-efficient 8-bit Atmel AVR microcontrollers. The reference platform for Marlin is the Arduino Mega 2560 with RAMPS 1.4, which is directly compatible with the equipment used for implementing the printer. This firmware runs on the mainboard and manages real-time controls for heaters, steppers, sensors, lights, LCD, buttons, etc. The control language is a derivative of G-code. G-code commands issue simple instructions, such as "set heater 1 to 180" or "move to XY at speed F".

The power supplies shipped with 3D printers usually generate up to 350 W on 12 V output. In our implementation, a more robust solution was preferred to accommodate the power requirements of the visual sensor. To this end, a 650-W ATX power supply was used. The electronics and the sensor are connected to the 5-V output while the stepper motors are connected to the 12-V output.

The prototype is shown in Figure 2. The rightmost image zooms into the attachment, which is mounted together with the printing head.

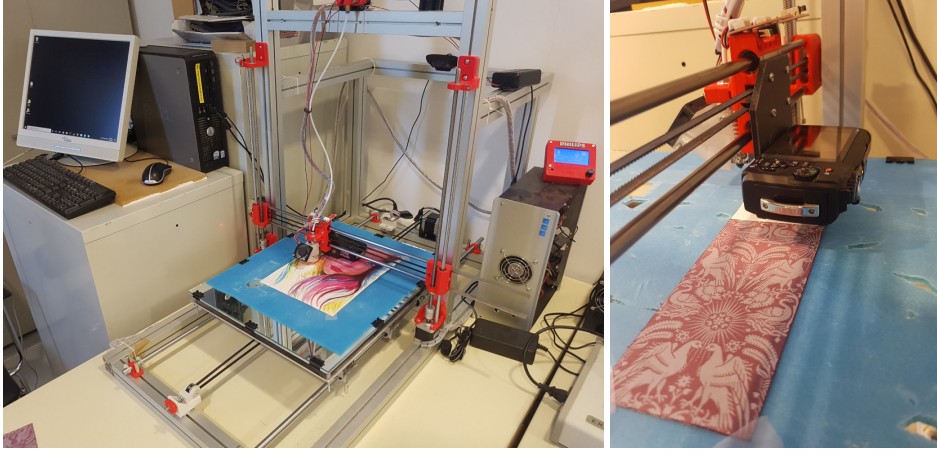

**Figure 2.** Prototype and close-up of the mounted sensor.

### 2.2. Motor and Motion

The implementation of a C3Dp contains some free variables, such as the quality of materials, the torque of motors, etc. To reduce the effect of vibrations and increase motion accuracy, the device was implemented as follows.

Aluminum frames of 40 × 40 mm thickness were used for the truss. The backside of the print bed was enhanced with an aluminum frame to increase its weight. High-quality, heat-hardened steel rods of 12 mm-thickness and high-quality linear bearings were used for the motorized part of the printing bed. Printable components of the apparatus were printed using PET-G and a 60% infill rate to enhance their stiffness and reduce the possibility of heating deformation due to intensive use.

To provide enough torque for this implementation, motors were standard Nema 17-sized high-torque stepper motors. The motor's motion was transmitted via 6 mm nonelastic timing belts, integrated with steel threads for enhanced stiffness. Motors are driven by the Texas Instruments DRV8825 Stepper Motor Controller ICs. The controller supports up to 1/32 microstepping. The device is operated through a microcontroller built on top of the Arduino Mega 2560. For the wiring of the C2Ds, the RAMPS 1.4 Arduino Mega Pololu Shield was used.

On account of the achieved mechanical robustness, the printing bed was increased by a factor of 4.764 from its specification to $50 \times 60$ cm$^2$. The bed was coated with a 5 mm-thick aluminum sheet to ensure a flat slide for the placement of samples.

### 2.3. Imaging

The camera faces the imaged surface perpendicularly. To mount the camera, a sensor base was designed using the TinkerCad software. The design, shown in Figure 3 (left), was exported in STL format and printed on the C3Dp—see Figure 3 (right). The design of the mount is compatible with the print head and is placed on its backside, allowing both heads to be mounted concurrently.

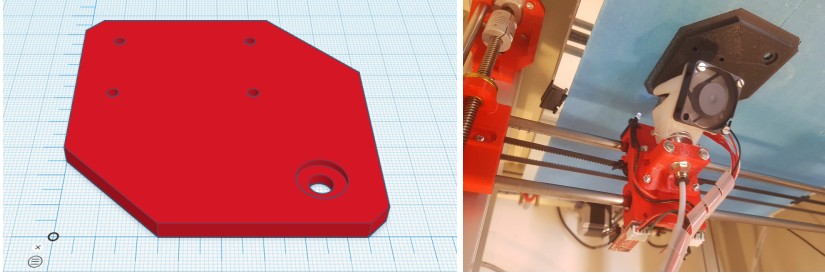

**Figure 3.** Design of the camera base (**left**) and photograph from its printing on the C3Dp (**right**).

The time required to capture the required number of images exceeds the duration of typical consumer-grade batteries, i.e., ≈1300 mAh. To avoid interrupting the scan for changing and the consequent sensor displacements, power was continuously provided as follows. A printed case emulating the battery was wired to a power supply of the appropriate voltage and current. The second component in Figure 4 guides and stabilizes wirings. The models can be found in the Supplementary Material.

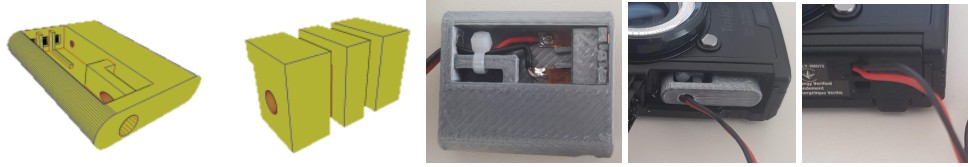

**Figure 4.** Designs of a battery emulator and photographs of its implementation.

### 2.4. Communication and Control

The scanner control software runs on a personal computer connected wirelessly to the controller. This computer runs software mediating the image acquisition process. The software executes a scanning plan, containing the locations of image acquisition, encoded as 3D coordinates.

Specifically, print bed and sensor are drawn to a designated relative position, due to a G-Code command to the controller, e.g., "G1 X2.57 Y1.93 Z18.20". The signal for image acquisition is then sent. The command sequence, encoded in HTTP, in Table 1 (top) is sent to trigger the acquisition of an image. Image acquisition failures and delays are treated as follows: The software checks if the image has indeed been acquired, using the HTTP command in Table 1 (bottom). This command acquires the list of stored images to be compared with a previously collected one. These two sequences are repeated until the image is acquired. Each filename is stored to later conveniently rename the acquired images.

When all the pictures are acquired, the files are manually copied from the memory card of the sensor and automatically renamed by the software. The renaming includes the coordinates of image acquisition in the filename in the form of Z-Y-X.jpg, such as '018.20-001.93-002.57.jpg'. These 3D coordinates are only readings of the C3Dp controllers, they are not regarded as absolute measurements but fused with visual cues, as discussed in Section 3.3.

**Table 1.** Sensor communication command sequences.

**Image Acquisition**

```
http://192.168.0.10/exec_shutter.cgi?com=1st2ndpush
http://192.168.0.10/exec_shutter.cgi?com=2nd1strelease
```

**File Listing**

```
http//192.168.0.10/get_imglist.cgi?DIR=/DCIM/100OLYMP
```

### 2.5. Cost

The cost of materials is reported in Table 2. On the date of submission, the total cost was 952 USD.

**Table 2.** Costs of utilized materials.

| Component | Quantity | Price (USD) |
|---|---:|---:|
| Visual sensor, Olympus Tough TG-5 | 1 | 430 |
| Microcontroller (Arduino Mega 2560) | 1 | 42 |
| Controller RAMPS 1.4 | 1 | 9 |
| DRV8825 Stepper Motor Driver | 5 | 20 |
| Stepper motors | 5 | 100 |
| Extruder | 1 | 10 |
| 30 mm $\times$ 30 mm aluminum truss | 3 m | 120 |
| Metal rods 12 mm | 2 pieces, 12 mm $\times$ 100 cm | 24 |
| Metal rods 10 mm | 4 cm $\times$ 60 cm | 28 |
| Aluminum sheet 5 mm | 2 pieces, 60 $\times$ 40 cm | 50 |
| Lead Screw T8 540 mm | 2 | 20 |
| Nut for Lead Screw T8 Lead 8 mm | 2 | 4 |
| Timing Belt XL 44″ | 1 | 10 |
| Aluminum GT2 Timing Pulley | 2 | 4 |
| Aluminum Flex Shaft Coupler 5–8 mm | 2 | 4 |
| Aluminum GT2 Timing Pulley Idler | 1 | 2 |
| ATX PSU 650 $W$ | 1 | 60 |
| Filament | 1/2 kg | 15 |
| **Total** | | **952** |

## 3. Method

### 3.1. Image Acquisition

Images are acquired in layers that form pyramids. For a pyramid layer, the locations of image acquisition form a hypothetical grid. These locations are determined so that images laterally overlap at all boundaries. For succeeding pyramid layers, overlap is medial. Camera centers are determined so that for consecutive layers, a node of the upper layer is a parent to the nodes of the lower layer that image the same surface region. To reduce scanning time and proprioceptive error, each layer is scanned in boustrophedon order.

Given the sensor's FoV and the proportions of lateral or medial overlap, camera locations are precomputed and stored in a tree-shaped data structure. These locations are depth-first serialized and converted to scanner coordinates. Images are acquired and indexed so that lateral and medial adjacency relations are retained.

### 3.2. Image Correspondence

Image correspondence is keypoint-based. We selected SIFT [18] as the baseline, but any other more suitable keypoint flavor can be used instead. Correspondences are sought in neighboring images, either laterally or medially. Matching accounts for the planarity of the imaged surface. Individual point matches are sought only within circular neighborhoods, as predicted by scanner motion. Correspondence establishment is symmetrical, as in [19]. Registration uses RANSAC [20] for robustness, using projective homography for the cost function. Correspondences are approved only if the reprojection error is below threshold $\tau$; otherwise, they are discarded.

### 3.3. Image Registration

We call a *map* the imaged surface, in pixel coordinates, in the coordinate frame of the mosaic to be created. The input to image registration is the proprioceptive estimates of the camera centers and the established point correspondences across laterally or medially adjacent images. The output is a set of projective homography transforms $H_i$, estimated for each image $I_i$, where $i$ enumerates the images across all pyramid layers. These homographies associate image locations in each $I_i$ to the corresponding locations in the mosaic.

World points $\mathbf{C}_i$ are the proprioceptively obtained coordinates for these locations in 3D space. Image points $\mathbf{c}_i$ are the image centers of images $I_i$. Initially, projective homography $H_g$ is estimated across this map and the 3D grid locations $\mathbf{C}_i$, using least-squares.

For each pair of adjacent images $I_i$ and $I_j$, we enumerate the correspondences between them using $k$ and denote their locations in $I_i$ and $I_j$ as $\mathbf{f}_{ki}$, $\mathbf{f}_{kj}$, respectively. The computation estimates the homographies by optimizing the following objective function:

$$\sum_i \sum_j (H_i \mathbf{f}_{ki} - H_j \mathbf{f}_{kj})^2 + \sum_i (H_g \mathbf{C}_i - H_i \mathbf{c}_i)^2. \tag{1}$$

The first term is the conventional reprojection error metric for point correspondences. In that term, $j$ enumerates the neighbors of $I_i$. The second term promotes compliance with the scanner coordinates. In Figure 5, the notation is illustrated.

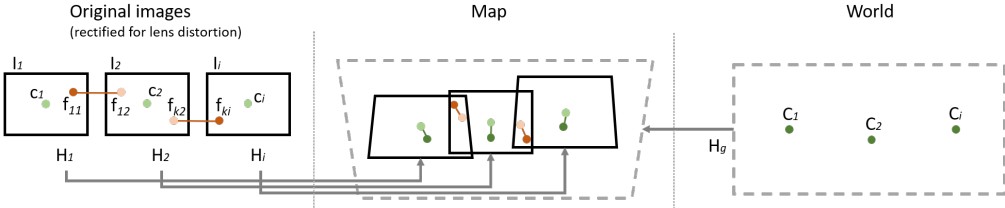

**Figure 5.** Illustration of objective function notation.

The projective homography has 8 free variables and, thus, the optimized variables are 8 times the number of images. The optimization capitalizes on the adjacency information

contained in the pyramid data structure, to create a topological graph, as the one in Figure 1 (right). This graph has points $\mathbf{C}_i$ as nodes and, as vertices, their adjacency relations. These relations constrain the search space of the optimization. We employed the work in [21], which is a framework for least-squares optimization of an error function that can be represented by a graph and has been specifically designed for SLAM or bundle adjustment problems.

An additional benefit of using the aforementioned graph-based method is the robustness to "missing estimates". Such a case was encountered in Section 3.2, where we discarded unreliable homography estimates. Another is the case where the apex of the pyramid cannot be reached by the hardware. The latter case is encountered when covering wider areas, using multiple, laterally overlapping pyramids. There is no special treatment for running the method in this way. The difference is the lack of the cues that would have been provided by images acquired from a larger distance.

### 3.4. Image Combination

The high definition of the macro lens at close distances comes at a significant cost, which is its shallow depth of field, or otherwise, the range at which imaged surfaces are in focus. This locus is a spherical shell centered at the focal point. For the macro lens, this shell is thin and small. The depth of focus is set representative of the fovea, such that the center of the image is best focused. As the imaged surfaces are planar, the image periphery is less focused. Another common issue in mosaics is the occurrence of seams at the stitching boundaries. Both issues are treated with the method in [22], applied for 32 spectral bands.

## 4. Results

The goal of the experiments was to assess mosaic registration accuracy and to explore tolerance to departures from the assumptions of Lambertian reflectance and the surface planarity. Indicative samples were drawn from applications relevant to sensitive materials, found in art, biology, document analysis, and textiles. The selected materials exhibit variability as to their reflectance properties and their 3D surface texture. We included shiny and rough materials in the samples. We did not include highly transparent, highly specular, or highly reflective materials.

Macroscopic images of the samples, acquired by a conventional camera, are shown in Figure 6. In the figure, from left to right and top to bottom, samples 1–5 are paintings; sample 6 is a blank piece of cotton canvas; sample 7 is a scarcely handwritten A4 page; sample 8 is a stamped and signed passport; samples 9 and 10 are blank and printed graph paper, respectively; samples 11–13 are pieces of silk fabric; sample 14 is a leaf; samples 15 and 16 are fine and coarse-grained sand, respectively; sample 17 is an assortment of coins; sample 18 is a banknote. Samples 13 and 18 were scanned entirely. For the rest, a $5 \times 5$ cm$^2$ region was scanned. In Figure 7, $2048 \times 2048$ p regions from images of the finest layer are shown, in the same order as in Figure 6.

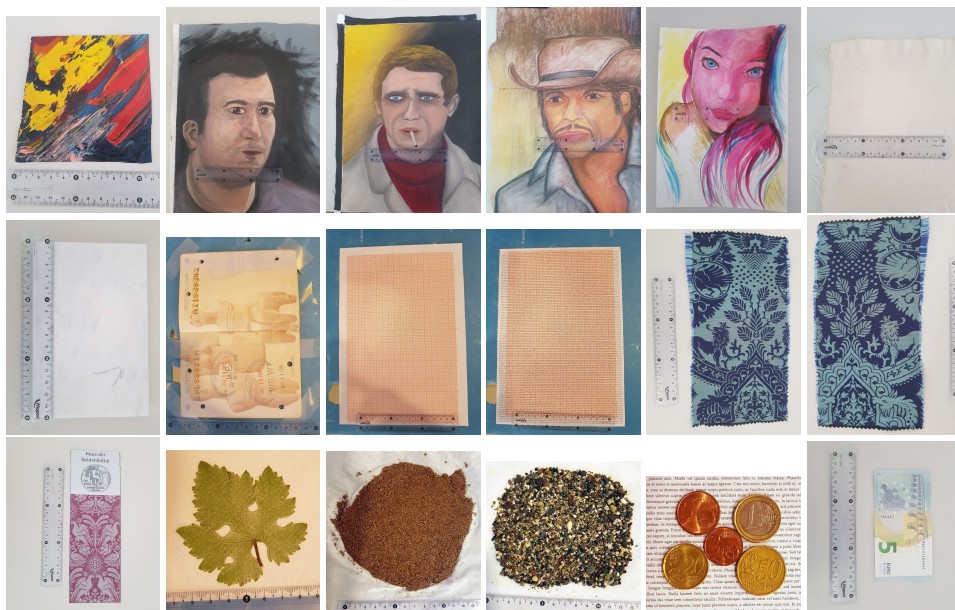

**Figure 6.** Samples.

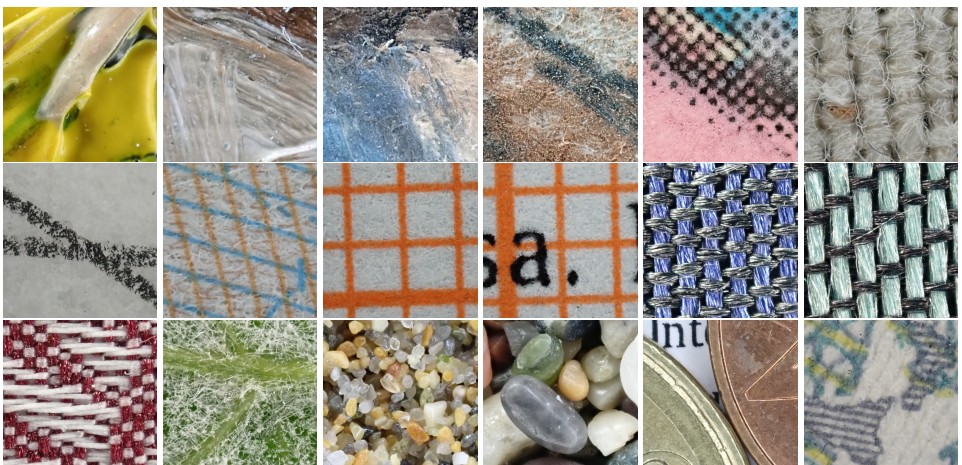

**Figure 7.** Sample details.

### 4.1. Conditions

In all experiments, the sensor was operated in autofocus and automatic stacking mode. The illumination was produced by the sensor's flash and ambient light. Sensor brightness, contrast, and color balance were set to automatic. The utilized sensor provides images encoded in JPEG format, at $4000 \times 3000$ p resolution. The average size of the image file is 2.4 Mb. The frequency of image reacquisitions, as per Section 2.4, was $\approx 1/1000$. The effective scanning area is $33 \times 37$ cm$^2$ and is a region of the printing bed at its center, allowing for laterally bounding paraphernalia.

The maximum elevation of the C3Dp was 30 cm. This elevation determines the height of a hypothetical pyramid. When the sensor is at that height, it occurs at its apex. In each layer of this pyramid, the lateral overlap is 50% for horizontal or vertical adjacency and 25% for diagonal. The pyramid has 5 layers, configured as in Table 3. Doubling the elevation per layer results in a medial overlap of $\approx 4$, meaning that a parent node fully overlaps with 4 images from a finer layer. This level of medial overlap was sufficient for the samples we scanned. Denser or sparser configurations are treated in the same way. For the utilized sensor, the base layer of this pyramid is $5 \times 5$ cm$^2$ and is covered by $25 \times 19 = 475$ images.

**Table 3.** Temporal and computational requirements.

| Layer | Elevation | X Steps # | Y Steps # | Start X | Start Y | Step X | Step Y |
|---|---|---|---|---|---|---|---|
| Fine 0 | 18.2 | 19 | 25 | 2.56 | 1.92 | 2.56 | 1.92 |
| 1 | 36.4 | 9 | 12 | 5.13 | 3.85 | 5.13 | 3.85 |
| 2 | 72.8 | 4 | 6 | 10.27 | 7.70 | 10.27 | 7.70 |
| 3 | 145.6 | 2 | 3 | 20.55 | 15.41 | 20.55 | 15.41 |
| Coarse 4 | 291.2 | 1 | 1 | 41.11 | 30.83 | 41.11 | 30.83 |

The inequality between the numbers of X and Y steps is by preference. Given the rectangular camera FoV, this configuration results in a square scanned area. The surface region covered by each image is shown in the top row of Figure 8. Each image corresponds to a pyramid layer, ordered from left to right and from coarse to fine scale. The surface area covered by each image is outlined using different colors for neighboring images. In the example, the square formed in the rightmost map is $5 \times 5$ cm$^2$. The bottom row shows the acquired images warped to the mosaic map or, in other words, the mosaics obtained for each pyramid layer. For the leftmost map, one image was warped; this image is the one acquired at the pyramid apex. The remainder maps are mosaics of warped images acquired at the locations denoted in the third and fourth column of Table 3. The rightmost mosaic comprises of 475 images. In the remainder of this section, the obtained mosaics are cropped to omit blank areas. The mosaics presented in Figures 9–13 image the same amount of area. The dimensions of the mosaics in Figures 14 and 15 are reported in Sections 4.2.5 and 4.2.7, respectively.

The sensitivity of the keypoint detector was tuned to its highest level. The number of keypoint features detected in the original images typically ranges 30–60 K features per image. Datasets with more intricate texture, such as the banknote and textiles, exhibited about 100 K features per image. However, the robust correspondences across image pairs are much fewer and are in the range of 0.5–5 K. The reprojection error threshold employed to judge the reliability of a homography estimate was $\tau = 25$ p (see Section 3.2).

The same computer was used in all experiments. Its specifications were as follows: CPU x64 Intel i7 8-core 3 GHz, RAM 64 Gb, GPU Nvidia GeForce GTX 4 Gb RAM (GTX1070), SSD 256 Gb, HDD 2 Tb. The critical parameter is RAM as it determines the number of correspondences that can be stored in memory and, ultimately, the number of images that can be stitched into a mosaic given the capacity of said memory. The use of time and computational resources, as a function of the scanned area, is reported in Table 4.

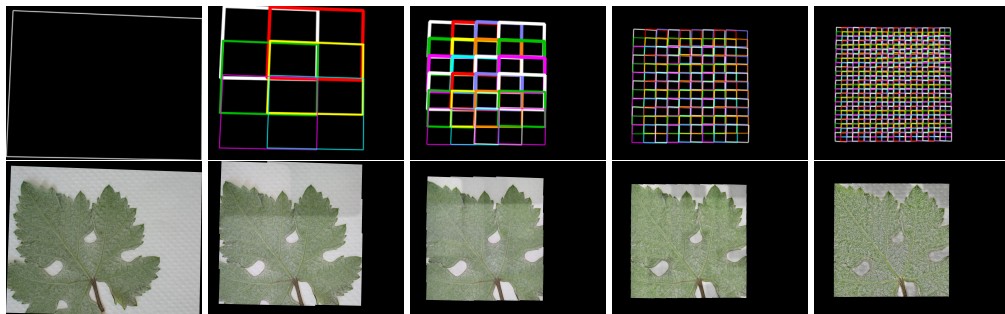

**Figure 8.** Scanned area and mosaics per pyramid layer.

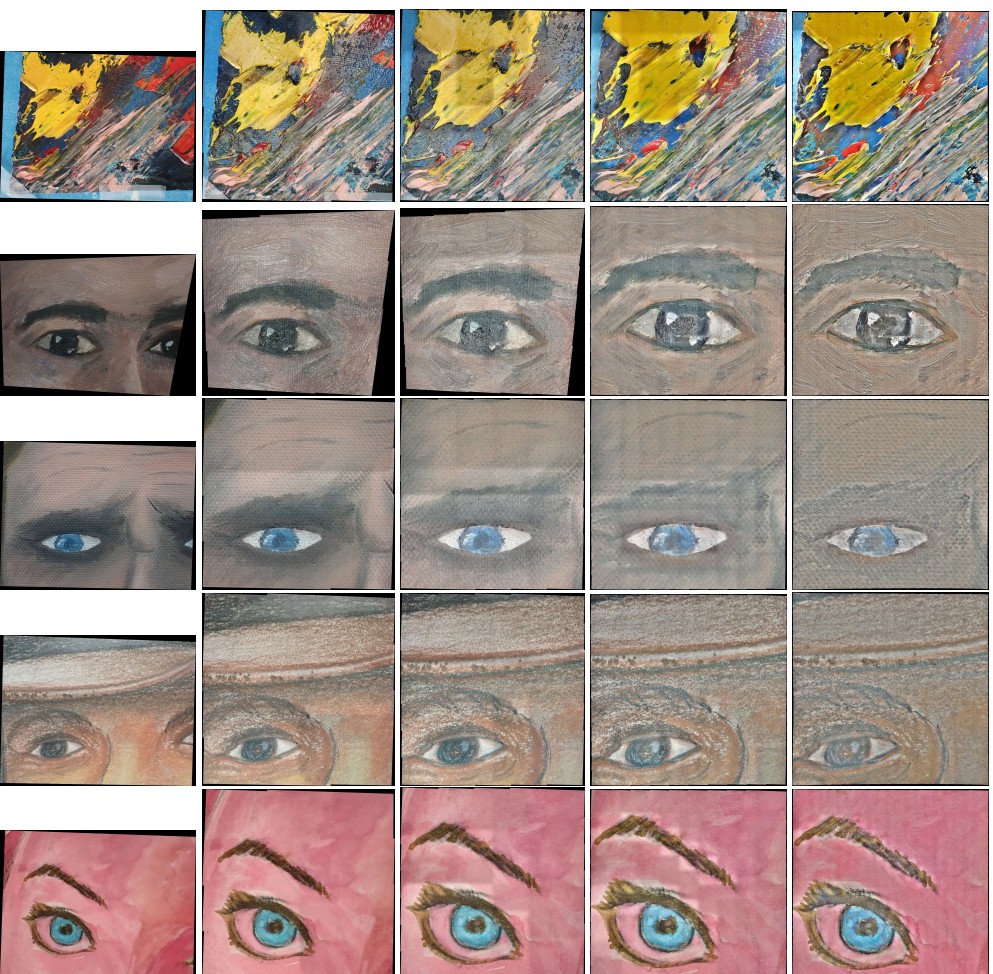

**Figure 9.** Paintings.

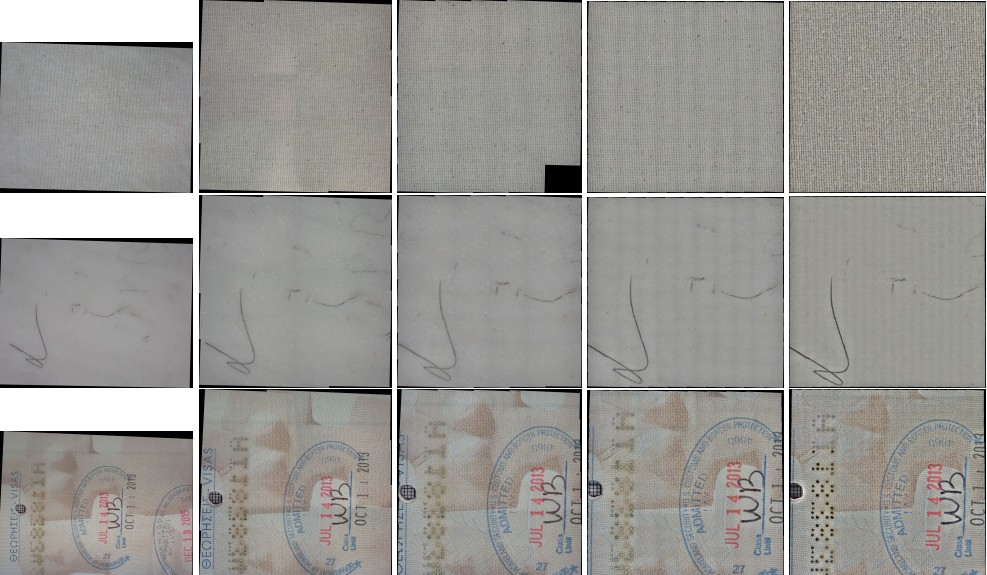

**Figure 10.** Canvas, handwritten paper, and passport with stamp and handwriting.

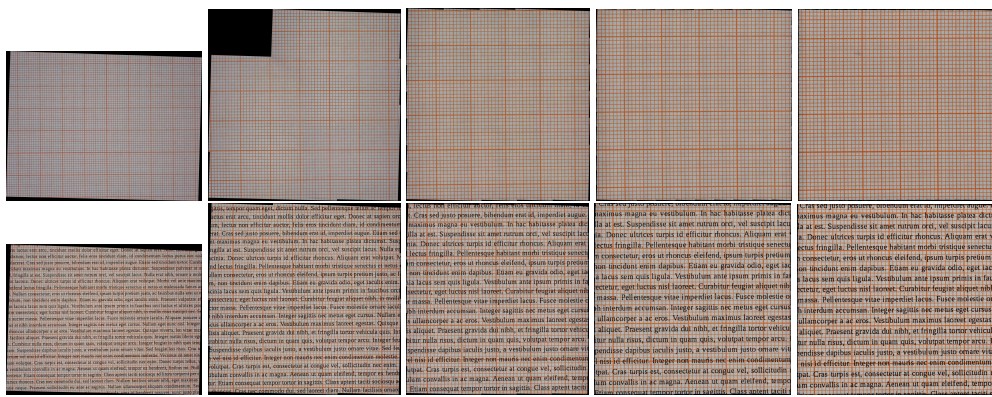

**Figure 11.** Repetitive patterns.

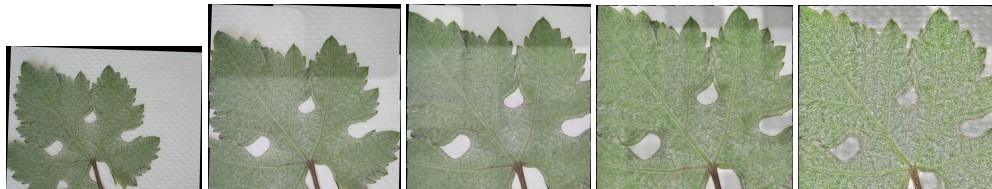

**Figure 12.** Fiona.

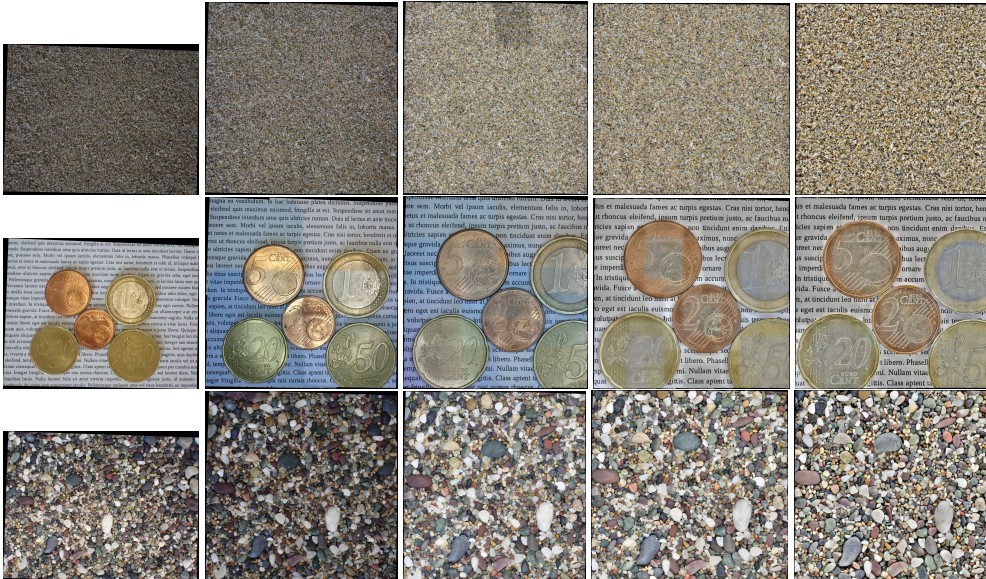

**Figure 13.** Approximately planar surfaces.

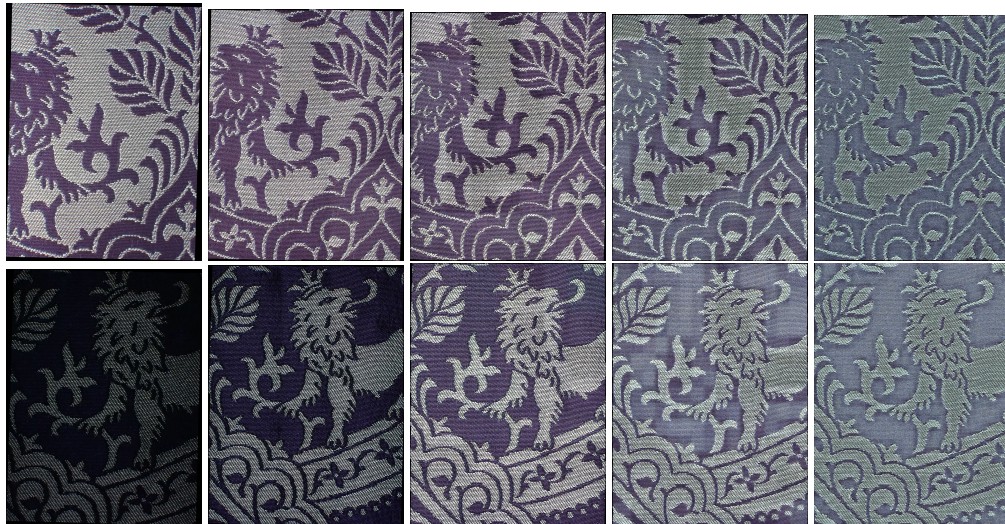

**Figure 14.** Fabrics.

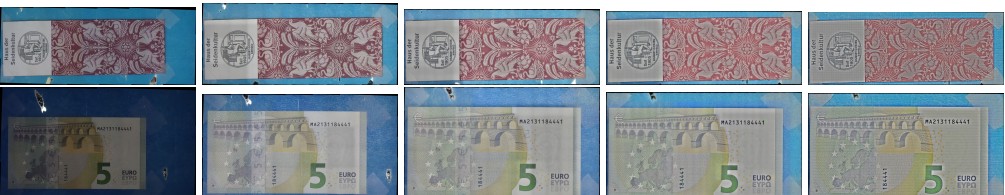

**Figure 15.** Larger scans.

**Table 4.** Temporal and computational requirements.

| Area (cm$^2$) | Scan time (h) | RAM (Gb) | Storage (Gb) | Computation Time (h) |
|---|---|---|---|---|
| 25 | 2 | 3.4 | 1.4 | 3.5 |
| 91 | 8 | 12 | 5.5 | 11.5 |
| 176 | 15.3 | 25 | 10.5 | 38.5 |

*4.2. Qualitative Tests*

4.2.1. Computational Behavior

Our primary investigation regarded the translational component of the estimated homographies. The estimated homographies lead to image shifts that are always less than 4 p. In other words, none of the estimated homographies suggest an update of $\mathbf{C}_i$ that would cause an image shift no larger than 4 p. In turn, this suggests that no homography estimate is in gross contradiction with the proprioceptive readings.

Second, we observe that the method is robust to the occurrences of missing information that were encountered in the experiments. The information missing was either a single pyramid apex in the use of multiple pyramids or registration failures due to a lack of reliable point correspondences. In all cases, a complete mosaic is provided for the layer of the highest detail.

4.2.2. Paintings

To study paintings, we acquired painted samples upon canvas, Canson paper, and regular printing paper. The samples exhibited various degrees of surface roughness. The colors were made from soft pastel or oil. The results are shown in Figure 9. To better investigate the effect of height variations, in the example of the top row, the impasto painting technique [23] (p. 100) was utilized. This technique involves painting in overlapping layers and gives rise to surface anomalies. The average height step of these anomalies was

0.75 mm. The example is centered upon a 1.5 mm protrusion. In the remaining rows of Figure 9, the examples are sorted in the declining level of surface roughness.

### 4.2.3. Paper and Canvas

A white sheet of paper and a white piece of canvas tested the application of the method in documents and plain fabrics. Canvas is a textile with repeated structure, but only macroscopically, as cotton fibers provide unique textures at the imaging resolution. Cotton plies are in the range of 12–20 µm. The results are shown in Figure 10. Under investigation were potential effects due to blank surface space. No such effects were observed, as paper when closely inspected reveals rich texture. The same was the case for cotton canvas used for painting. Still, in the third layer from the top, a homography estimate was discarded.

### 4.2.4. Repetitive Patterns

To test against the sensitivity of feature-based image registration to repetitive patterns, conventional graph paper was used. The results are shown in Figure 11. As in the case of the canvas, in the fine mosaic layers, sufficient uniqueness cues are found to abstain from gross misregistration errors. Nevertheless, a failure is observed in the autofocus function of the sensor. At the second layer from the top, some images were out of focus, possibly due to the sensitivity of the autofocus mechanism of the sensor to repeated texture. The homography estimate was considered unreliable and, thus, discarded. We repeated the experiment this time using printed text, using the "Liberation Serif —Regular" font, at 6 pt. In this condition, said effect did not occur.

### 4.2.5. Fabrics

Though the study of fabrics is related to heritage [24] and industrial applications, ways to digitize textiles and fabrics are constrained in the products reviewed in Appendix B. These approaches do not scan the fabric in sufficient resolution to reveal the fine crafting of some textiles. We chose patterned silk fabrics handwoven on a Jacquard loom because this type of weaving allows for intricate patterns on the fabric. We chose silk as the most challenging material, because its fibrils are sleek, reflecting light from many angles, attributing it with its characteristic sheen. Moreover, silk is one of the finest plies. Silk fibers from the Bombyx Mori, as in the example, are in the range of 5–10 µm (a human hair is $\approx$ 50 µm). We scanned two samples woven with the same two-color pattern but with alternating colors. The results are shown in Figure 14.

### 4.2.6. Fiona

To image biological tissue, a leaf was scanned. The result is shown in Figure 12. Biological samples avail more information when backlit. If illumination frequency is modulated, spectral absorption measurements can be obtained. We plan such a version of this scanner for the future.

### 4.2.7. Larger Scans

We tested the utilization of multiple, laterally overlapping pyramids in samples of larger areas. The targets were a $21 \times 7$ cm$^2$ piece of industrially woven, patterned silk fabric and a $12 \times 6.2$ cm$^2$ banknote. The results are shown in Figure 15. The pyramids used were $X$ and $Y$ for these cases, respectively. For the fabric, an arrangement of $5 \times 2$ pyramids was used and, thus, the top layer was a mosaic of 10 images. For the banknote, a $2 \times 3$ arrangement was used and, thus, the top layer comprised 6 images. In this configuration of RAM and scanning resolutions, the maximum scan size is $42 \times 14 = 588$ cm$^2$. In the Supplementary Material, the scans of both sides of the banknote are provided.

### 4.3. Benchmark and Quantitative Experiments

Quantitative experiments were challenging due to our inability to accurately manufacture targets of known size and with local features at the fine scale required. For this reason, we used structures of known size, such as coins and security prints.

#### 4.3.1. Resolution

We report an optical resolution of 19,754 horizontal and 19,820 vertical ppi, without interpolation. Thus, mosaic pixels deviate by a factor of $\phi = 0.00334$ from squareness. The measurement was obtained using the banknote, which is of known dimensions. This means that mosaics are linearly scaled by a factor of $\phi$ in the vertical direction. If needed, the mosaic can be resampled to feature square pixels. We report the horizontal as the scanner resolution—19,754 ppi. The benchmark was obtained using a banknote and *mm*-grade graph paper.

#### 4.3.2. Mosaic Encoding

For a $5 \times 5$ cm$^2$ mosaic with 5 layers, 626 images were acquired, as described in Table 4. Their storage capacity is $\approx$1.5 Gb. As the mosaic images are overly large for conventional image viewers, we render them in 256 p $\times$ 256 p image partitions called "tiles". In the Supplementary Material, tiles are provided in their original resolution encoded in JPEG format. The resolution of mosaics and their storage requirements are shown in Table 5. In the Supplementary Material, a hierarchical viewer (OpenSeadragon 2.4.2) is provided that enables the inspection of all layers in their original resolution, similarly to photorealistic mapping systems.

**Table 5.** Mosaic resolution and storage capacity in JPEG encoding.

| Layer # | Resolution (Gp) | Storage (Mb) |
|---|---:|---:|
| 0 | 3707.8 | 395 |
| 1 | 927.0 | 97 |
| 2 | 23.7 | 27 |
| 3 | 56.0 | 9 |
| 4 | 14.5 | 3 |

To be easily accessed, mosaics are rendered in smaller resolutions, such as in the figures of this document. In the Supplementary Material, the mosaics for each layer are provided in their original resolution.

#### 4.3.3. Approximately Planar Surfaces

To observe the effects of surface roughness, we scanned two types of sand and an assortment of coins. The results are shown in Figure 13, in increasing order of surface roughness. The coins ranged from 1.67–2.33 mm in thickness. For the coarse-grained sand, height steps of grains between adjacent grains well-exceeded 3 mm.

We did not detect artifacts for fine-grained sand or coins. Though not easily found, the case of coarse-grained sand exhibits some tractable mismatches, as the image combination method cannot compensate for the lack of accurate registration between the overlapping image regions. They are shown in Figure 16, where each image shows a region of $\approx$1 cm$^2$.

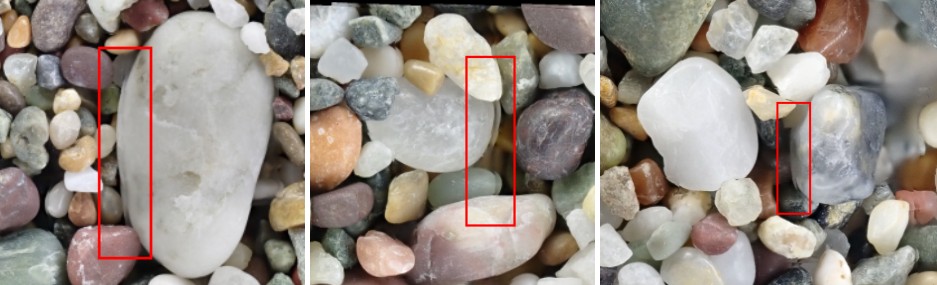

**Figure 16.** Registration failure examples.

### 4.3.4. Image Structure

Ideally, when viewed at the same resolution, mosaic layers that image the same surface region should be identical. The similarity between mosaic layers was quantified by cross-correlation in the domain of [−1, 1]. We computed this metric between consecutive layers, as well as between the top and bottom layer. In Table 6, we report the correlation values. The first row shows the correlation coefficient between the coarsest and the finest layer. The remainder columns show the correlation values for consecutive layers.

**Table 6.** Correlation coefficients between mosaic layers.

|  | 1 | 2 | 3 | 4 | 5 | 6 | 7 | 8 | 9 | 10 | 11 | 12 | 13 | 14 | 15 | 16 | 17 | 18 |
|---|---|---|---|---|---|---|---|---|---|---|---|---|---|---|---|---|---|---|
| **0–4** | 0.5 | 0.3 | 0.1 | 0.2 | 0.6 | 0.6 | 0.5 | 0.6 | 0.9 | 0.9 | 0.2 | 0.5 | 0.6 | 0.3 | 0.7 | 0.8 | 0.3 | 0.5 |
| **0–1** | 0.9 | 0.8 | 0.9 | 0.9 | 0.9 | 0.8 | 0.7 | 0.8 | 0.5 | 1.0 | 0.9 | 0.8 | 0.9 | .9 | 0.9 | 0.9 | 0.5 | 0.8 |
| **1–2** | 0.8 | 0.7 | 0.7 | 0.9 | 0.9 | 0.4 | 0.6 | 0.9 | 0.5 | 1.0 | 0.9 | 0.9 | 0.9 | 0.9 | 0.9 | 0.9 | 0.6 | 0.9 |
| **2–3** | 0.6 | 0.3 | 0.6 | .8 | 0.9 | 0.5 | 0.6 | 0.9 | 1.0 | 1.0 | 0.8 | 0.8 | 0.9 | 0.8 | 0.9 | 0.9 | 0.6 | 0.9 |
| **3–4** | 0.7 | 0.5 | 0.4 | 0.4 | 0.8 | 0.7 | 0.7 | 0.7 | 0.9 | 1.0 | 0.2 | 0.7 | 0.8 | 0.7 | 0.8 | 0.8 | 0.7 | 0.9 |

To measure systematic distortions, we used the images of coins (from Section 4.3.3), which are circular structures. We performed Canny edge detection [25] in the finest mosaic layer and selected the edges corresponding to the circular creases of the 2 c and a 5 c coin. The selected edges were used to fit circles, using least-squares without a robust selection of inliers. In Figure 17, the selected edges are shown on the left pair of images. In Table 7, we report deviations of the detected edges from the fitted circle.

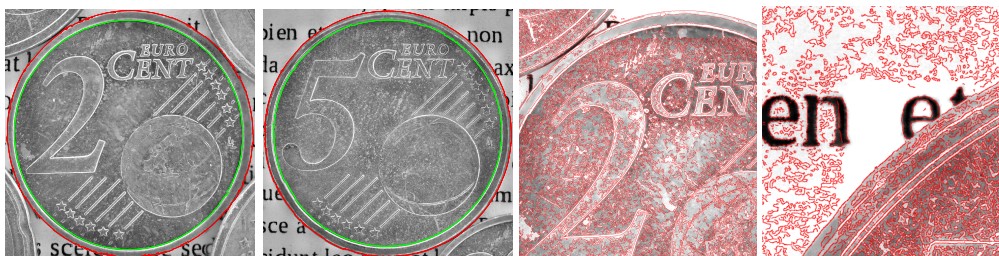

**Figure 17.** Edges belonging to the inner and outer creases of two coins (**left**) and edge detection details (**right**).

**Table 7.** Mean circle fit error and standard deviation.

| Circle | Radius (p) | Error (p) |
|---|---|---|
| 2 c outer | 3464.8 | 3.05 (2.62) |
| 2 c inner | 3216.8 | 4.09 (3.50) |
| 5 c outer | 3936.4 | 4.86 (3.17) |
| 5 c inner | 3630.8 | 5.20 (3.36) |

This example facilitates observations at the locations of lateral image overlap, where seams are typically observed. In general, due to accurate registration and the effect of the method in [22], seams are usually easily observable in fine scales. In the experiment, the focus distance was automatically adjusted. As the coins are elevated from the background, near the coin boundary, the sensor focuses either on the paper background or the coin. When the focus is placed on the background, the image region where the coin appears is blurry and, instead, the background is focused. Though the structure distortion is minute, the difference in the focus of the blended images is observable, when image edges are detected as in the right pair of images in Figure 17. In turn, the different amounts of image blur at the boundaries of the blended image gives rise to edge dislocations. In Figure 18, a mosaic of 2 × 2 images is shown for each sample in the experiments (minified for document scale). In the Supplementary Material, these images can be found in their true resolution. We observe that although no high frequent seams are observed, global brightness difference is observed between stitched images.

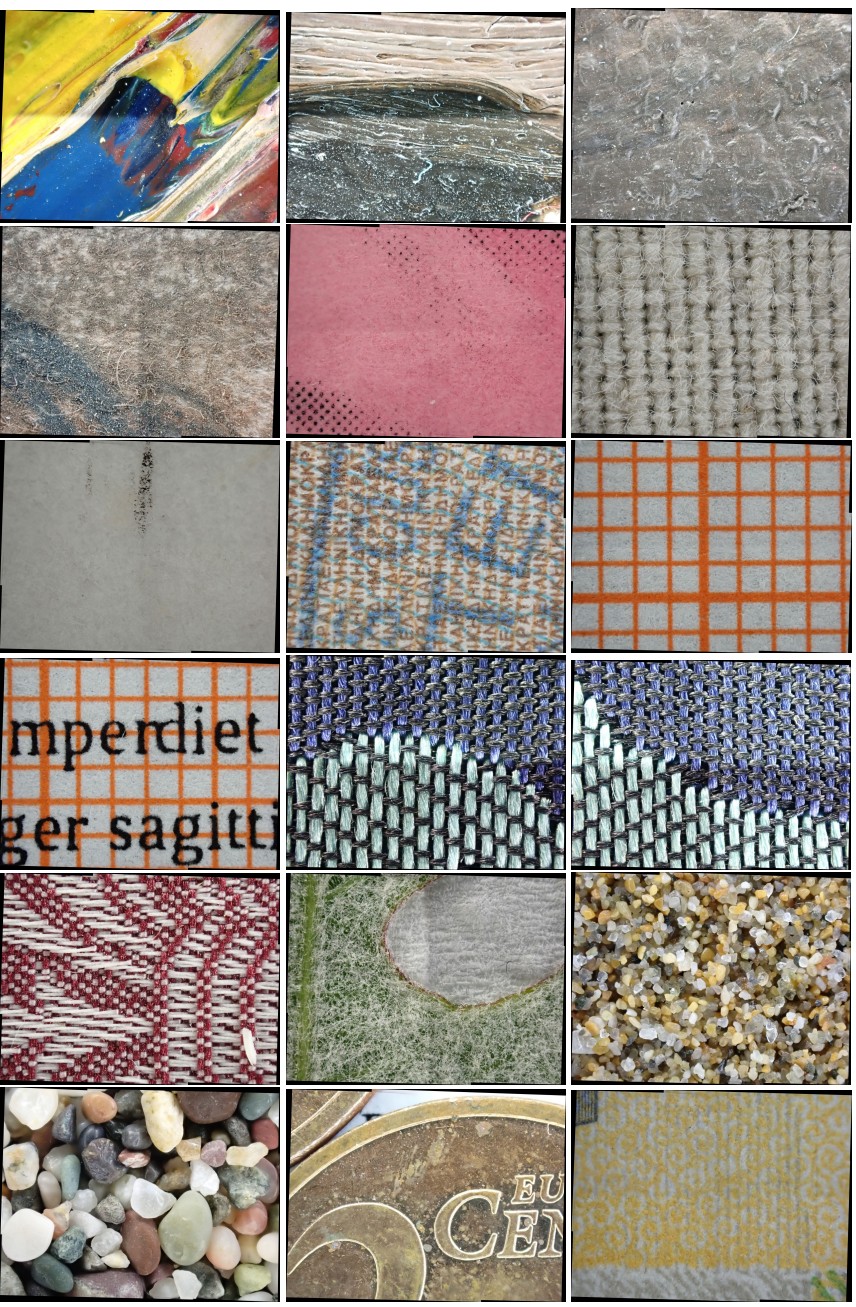

**Figure 18.** Mosaics of 2 × 2 original images.

## 5. Discussion

The quantitative experiments show that registration errors are in the order of 10 p in mosaics comprising $\approx$ 4 Tp. Correlation across layers was consistently positive, thus pointing to the nonoccurrence of gross registration errors. We also found no evidence of systematic distortion or error accumulation.

The obtained mosaics were inspected for distortions due to departures from the planarity assumption. We found the limitation of the current configuration to be sharp steps of over 3 mm.

We performed no correction as to the global optimization of image intensities. Setting the camera acquisition mode to automatic brightness adjustment adapts the dynamic range of image acquisition to the content of each image. This can be observed in the mosaics of the top layers, where brightness differences across surfaces of the same luminance can be observed. Compensation methods tailored for mosaics exist in the literature, e.g., [26,27]. On the other hand, reflectance calibration even by simple means, i.e., "gray card", supports the veridical measurement of lightness. It remains to be studied whether images of higher dynamic range are required to capture the brightness variations encountered in all images.

We did not control the stacking process provided by the sensor. By assigning this control to the embedded system accompanying the sensor, we may be wasting potential sensitivity to depth variations. Control of bracketing techniques would provide better focus and, thus, more image features. In addition, it can be supported even by weak depth cues, such as depth from defocusing [28] or stereo vision.

There are two main factors relevant to scaling the proposed approach for larger scans: the size of the C3Dp bed and memory of the computer that runs the optimization, as per Section 3.3. Larger setups can be achieved using open-source platforms, such as the MPCNC, to more precisely control motion over areas up to $2 \times 2$ m$^2$.

## 6. Conclusions

A surface scanning approach and its implementation are proposed in the form of a scanner imaging modality. The proposed approach employs auxiliary images to strengthen image registration cues and fuses proprioceptive data to produce mosaics of the scanned surface with a resolution of 19.8 Kppi. We conclude that the resultant device and approach offers a useful imaging modality for several applications in a cost-efficient manner.

**Supplementary Materials:** All of the mosaic results shown in this paper can be found in their original resolution at http://doi.org/10.5281/zenodo.4983052, accessed on 3 July 2021. A supplementary video presentation of two indicative results can be found in the supplementary material and is also available online at https://youtu.be/3rhcAUcekkM, accessed on 3 July 2021.

**Author Contributions:** Conceptualization, X.Z. and N.P.; methodology, X.Z., P.K., N.P. and N.S.; software, P.K. and N.S.; validation, X.Z., P.K. and N.S.; resources, N.P.; data curation, X.Z. and N.P.; writing—original draft preparation, X.Z. and N.P.; writing—review and editing, X.Z. and N.P.; visualization, P.K. and N.P.; supervision, X.Z.; project administration, X.Z.; funding acquisition, X.Z. and N.P. All authors have read and agreed to the published version of the manuscript.

**Funding:** This work was supported by the European Commission Horizon 2020 Project, Mingei, Grant No. 822336.

**Institutional Review Board Statement:** Not applicable.

**Informed Consent Statement:** Not applicable.

**Data Availability Statement:** All of the results presented in this work can be found at http://doi.org/10.5281/zenodo.4983052, accessed on 3 July 2021. The original images utilised as input data to produce the results, can be found at http://doi.org/10.5281/zenodo.5012566, accessed on 3 July 2021.

**Acknowledgments:** The authors thank Haus der Seidenkultur for the provision of patterned silk fabric samples and the anonymous reviewers for constructive criticism that led to the improvement of this manuscript.

**Conflicts of Interest:** The authors declare no conflict of interest.

## Abbreviations

The following abbreviations are used in this manuscript:

| | |
|---|---|
| C3Dp | Cartesian 3D printer |
| C2Ds | Cartesian 2D scanner |
| FoV | Field of View |
| GPS | Global Positioning System |
| HTTP | Hypertext Transfer Protocol |
| JPEG | Joint Photographic Experts Group |
| USD | Unites States Dollar |
| mAh | milliamp Hour |
| c | cent |
| G | giga |
| h | hours |
| m | millimeter |
| μ m | micrometer |
| cm | centimeter |
| kg | kilogram |
| p | pixel |
| pt | point |
| ppi | points per inch |
| T | tera |
| V | Volt |
| W | Watt |

## Appendix A. Printed Components

**Table A1.** Printed components. Accessed on 3 July 2021.

| Name | Source |
|---|---|
| Z-axis leadscrew | https://www.thingiverse.com/thing:519391 |
| Controller case | https://www.thingiverse.com/thing:2047732 |
| Bowden extruder | https://www.thingiverse.com/thing:2243325 |
| x-carriage | https://www.thingiverse.com/thing:2514659 |
| z-axis | https://www.thingiverse.com/thing:1692666 |
| y-axis belt holder | https://www.thingiverse.com/thing:1030200 |
| y-belt tensioner | https://www.thingiverse.com/thing:3404464 |
| y-axis motor holder | https://www.thingiverse.com/thing:2808408 |

## Appendix B. Market Survey

All prices are approximate and estimated on the day of submission. Unreported prices are ones that require asking for a quote and all exhibit a larger price than the others in the same table.

**Table A2.** Flatbed A0 scanners.

| Name | Optical Resolution (ppi) | Price (USD) |
|---|---|---|
| Kurabo K-IS-A0FW | 1000 | 50 K |
| Microtek LS-4600 | 600 | 60 K |

**Table A3.** Film scanners.

| Name | Optical Resolution (ppi) | Price (USD) |
|---|---|---|
| Plustek OpticFilm 8100 | 7200 | 400 |
| Epson Perfection V550 Photo | 12,800 | 600 |
| UScan+ HD LTE | 2400 | Quote |

**Table A4.** Large format scanners.

| Name | Optical Resolution (ppi) | Price (USD) |
|---|---|---|
| Colortrac SmartLF Sci | 1200 | 5–12 K |
| Colortrac SmartLF SC 42 Xpress | 1200 | 7 K |
| Contex IQ Quattro X - 44" | 1200 | 7 K |
| Contex IQ Quattro 4450/4490 | 1200 | 7–8 K |
| Image Access WideTEK 48CL | 1200 | 6 K |
| ROWE 850i - 55" | 1200 | 22 K |
| Image Access WideTEK 60CL | 1200 | 12 K |
| CRUSE ST Light 300 | 300 | 30 K |
| CRUSE ST Light 600 | 600 | 30 K |
| CRUSE Synchron Table (ST) | 830 | 30 K |
| CRUSE CS 85/145 ST-T | 600 | 30 K |
| CRUSE CS 82 ST-T 2450 | 600 | 30 K |

**Table A5.** Book scanners.

| Name | Optical Resolution (ppi) | Price (USD) |
|---|---|---|
| Suprascan double A0 | 600 | Quote |
| Suprascan Quartz A0 LED HD | 600 | Quote |
| Sma Scanmaster 0 3650 | 600 | Quote |
| book2net Hornet | 400 | Quote |
| Czur ET16 | 275 | 355 |
| Fujitsu Scansnap SV600 | 280 | 500 |
| SMA ScanMaster 2 | 1200 | 10 K |
| SMA RoboScan 2 | 600 | 10 K |
| IID Bookeye 5 V3 | 600 | 10K |
| Bookeye 4 V3 Kiosk | 600 | 10 K |
| Bookeye 4 V2 Semiautomatic | 600 | 10 K |
| Bookeye 4 V2 Professional Archive | 600 | 10 K |
| Zeutschel OS Q1 | 600 | 10 K |
| Zeutschel OS HQ | 1000 | 10 K |
| Zeutschel OS Q0 | 600 | 10 K |

**Table A6.** Material scanners.

| Name | Optical Resolution (ppi) | Price (USD) |
|---|---|---|
| Vizoo xTex | 1000 | Quote |
| xrite TAC7 | 385 | Quote |

**Table A7.** Scanning services.

| Name | Optical Resolution (ppi) | Price (USD per Sample) |
|---|---|---|
| Materialcapture | 600 | 100–300 |
| Muravision | 920 | 100–300 |
| Overnight scanning | 600 | 100–300 |

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
