# Peer review of "A Low-Cost Contactless Overhead Micrometer Surface Scanner"

_applsci, doi:10.3390/app11146274_

Round 1

Reviewer 1 Report

The paper present a interesting low cost solution for multi-resolution scanner. The results seems interesting and promising if following flaws (on the way to describe the data) can be corrected:

  1. All the images data author presented from page 9 to page 18 are not properly labeled. I cannot figure out the thumbnail image and 25% optical resolution image are a single tile image captured at moderate and close distance with middle and high resolution or stitched image from multiple tiles. The former cannot support the paper's conclusion. If it is later case, the author should give details on how many tiles used for the stitching. It looks like only two tiles are used in the 25% optical resolution stitching from the sand images in page 14. Such tile number is too few to prove the results, the author should show a 3X3 or at least 2X2 image to show the stitching effect.
  2. Low cost is major feature claimed in the paper. However, it seems they not include all elements in the table such as slide stroke and motor shaft etc, which are not ignorable on the cost. Also maybe they should include the partial cost of Cartesian 3D printer to make the comparison more fare.
  3. The paper should be carefully revised to make it approach the basic standard of the journal paper. Not only the grammar and typo error but also some issues like all pic should have scale bar (not only the first one with some hard to see ruler photo) and all picture has proper caption for the reader to identify them in the text undoubtedly.

Overall, the paper presents a interesting and maybe (due to its confused presentation in current draft) promising solution for multi-scale and multi-resolution scanner. The major weakness is its very limited scanning area 5cmx5cm or at most 40cmX4cm. However, besides that limit, the writing of the paper need a major revise before it can be accepted for publication.

Reviewer 2 Report

This paper refers to the design and implementation of a low-cost contactless scanner and its software. The specific subject has a great interest in many applications.

The journal’s template is modified (line numbers do not exist) and it is very difficult to review it.

The manuscript needs editing for language.

There is no consistency between American English and UK English (e.g. micrometer in the title and micrometre in the abstract).

The current manuscript has flaws in its justification of the research direction and method. The authors are strongly recommended to include a literature review on the existing technologies, which would help the justification of the novelty of the method (besides the cost).

The authors mention that they evaluate qualitatively and quantitatively the scanner but the “quantitatively” section is not clear.

The conclusion section is still not really presenting a drawn conclusion.

According to the instructions for authors, “References must be numbered in order of appearance in the text”. Also, there is no consistency with reference numbers in the text and the references (numbers 10, 11, 20, 21, 22, 30 are missing in the text).

The are many more comments but it is impossible to mark them without line numbers.

Reviewer 3 Report

Review

In the manuscript, the authors present a contactless scanner design that can be produced much cheaper than devices with similar resolutions currently available on the market, even in a smaller laboratory environment.

Regarding the title of the manuscript, I would definitely like to note that I do not consider the term “Poor-man’s” to be appropriate in terms of technical content, despite the fact that the authors specifically mentioned and provided the reader with an explanation of the choice of term. I suggest using a synonym that fits the appropriate context in the title of the manuscript (low-cost, cost-effective or cost-saving).

The practical applicability and wide applicability of the development presented in the manuscript is not in question, because a low-budget tool is obviously a welcome solution in a market environment that offers tools that are otherwise orders of magnitude more expensive.

In the Introduction section, the authors present the challenges and possibilities related to the scanning of flat surfaces, the solutions that can be implemented by overlapping, and the presence and effect of distortions and scanning errors that occur during scanning.

The authors present the scanning features and limitations of proprioceptive image registration, flatbed scanners, large-format scanners, and book scanners.

In the Material section, the authors present their self-developed interface scanner device, along with an accurate description of the part, subassembly, operating functions, communication, and control aspects.

The Method section provides a detailed description of the scanning process, matrix scanning, image combination, overlap handling, and the lens and optical sensor used to maintain the quality of the scan layers.

The Results section summarizes the environment designed for testing the scanner device, analyzing in detail the in-process temporary and final image storage resource needs. The authors present the surface samples used during testing (painting, textiles, paper money, coins, biological sample, fabrics, etc.) and illustrate the results of scanning and imaging by scanning layers. An examination of performance and its results in terms of resolution, image structure, and the correctness of the image registration process for repetitive samples are presented. A separate study was also performed on the scan of only approximately flat surfaces.

The Discussion section summarizes the results and makes observations about the limitations of the physical size of the scanner. In addition, the Conclusions section states that the scan gives satisfactory results, in addition to a cost-effective solution, but there are also a number of engineering and computational tasks that are worth researching and developing on the device.

The description of the development, the testing process, and the results achieved are clear, well usable, and are expected to be of great interest.

The scope, structure, and language of the manuscript correspond to the standard of publication related to scientific research and development.

The amount of literature used in the preparation of the manuscript is adequate, the referenced literature is related to the current topic discussed.

-  missing explanation on the first use of the abbreviation C3Dp (on 3rd page )

- typing error: on 3rd page „frontal posture of te volume”->” frontal posture of the volume”

- typing error: on 3rd page „pertain only to inrement” -> „pertain only to increment”

-  What kind of micro-controller can be used to control the device? Please give additional information about the used micro-controller earlier (it was mentioned at the bottom of 3rd page, but was defined later, on the 4th page)

- There is a brief description of the power supply on the 5th page. What is the minimum and what is the optimal capacity of the usable power supply in this construction? Is it different from the 3D printer’s original power supply? On which parameters depend the selection of the power supply?

- on the 5th page it was mentioned: „However, image acquisition failures and delays are part of the mechanic problem to solve.” Is it solved or is it a further task during the development?

-  Acquired files’ names contain X-Y-Z coordinates. The transfer from the memory card is made manually, but the renaming process is automated. What ensures that the file name contains the correct coordinate data?

- Question to this statement (on 6th page): „camera locations are precomputed and stored in a tree-shaped data structure” - Does the inaccuracy due to the physical nature of the stepper motor not negatively affect the image overlapping and image registration?

- The following texts are not required, I suggest deleting it: „Let i enumerate the set of all images Ii. Image points ci are their image centres and world points Ci the locations of these centers in 3D space.” and „Let fki be the kth feature in Ii.” As there is no detailed formal description in the manuscript, no formal definition is required in the text. In the following, I suggest placing them in Figure 5.  as an explanation of the marking or place them in the 3.3 subsection.

- please give more detail and explanation about the optimization method (mentioned at the bottom of the 6th page). The „using[32]” is not enough explanation.

- Results section: near this content „Results are presented in three columns. The left shows...” please refer to the concrete figure instead of „left”. E.q. „…presented in Figure X. Please add figure numbering and figure captions to the three-column figures.

- 7th page: please clarify the measure units in the 4. subsection, because it is hard to decide what does the mm2 means. Maybe a quadratic or a cubic unit?

- 8th page:  Please introduce the main attributes of the computer that was used to perform a computational task.

Great work, congratulate the Authors.

Round 2

Reviewer 1 Report

The revised version has successfully addressed most of my comments and i will support its publication

Reviewer 2 Report

Satisfied with the corrections.